ecology

plant communities, vegetation structure, plant traits, Verloren Valei nature reserve, Middelpunt Wetland, Lakenvlei wetland

**Author for correspondence:**
L. R. Brown
e-mail: lrbrown@unisa.ac.za

# A vegetation classification and description of white-winged flufftail (*Sarothrura ayresi*) habitat at selected high-altitude peatlands in South Africa

A. J. Marais[1,2], K. Lloyd[3,4], H. A. Smit-Robinson[2,3] and L. R. Brown[2]

[1]Aquatic Unit Lydenburg, Mpumalanga Tourism and Parks Agency, Postnet Suite #4 P/B X20097, Lydenburg 1120, South Africa
[2]Applied Behavioural Ecology and Ecosystem Research Unit, University of South Africa, Private Bag X6, Florida 1710, South Africa
[3]Conservation Division, BirdLife South Africa, Private Bag X16, Pinegowrie 2123, Gauteng, South Africa
[4]Department of Statistical Sciences, University of Cape Town, Rondebosch 7700, South Africa

LRB, 0000-0002-1026-5438

The white-winged flufftail is listed as critically endangered, and limited knowledge about the species' ecology has been identified as a limiting factor to effectively conserving the bird. Little is known about the vegetation inhabited by the white-winged flufftail, which hampers the identification and management of its habitat. This study presents a fine-scale classification and description of the vegetation of wetland sites where the bird is known to be present. A plant phytosociological study was conducted to describe the plant communities and vegetation structure of the habitat. Three sites were selected at Verloren Valei Nature Reserve and two at Middelpunt Wetland, Mpumalanga, South Africa, shortly after the white-winged flufftail breeding season. A total of 60 sample plots were placed within the study sites, where all plant species present were recorded and identified. Other aspects such as plant height, water depth and anthropogenic influences were also documented. A modified TWINSPAN analysis resulted in the identification of three sub-communities that can be grouped into one major community. The Cyperaceae, Asteraceae and Poaceae families dominate the vegetation, with the sedges *Carex austro-africana* and *Cyperus denudatus* being dominant, and the grasses

*Leersia hexandra* and *Arundinella nepalensis* co-dominant. The broad habitat structure consisted of medium to tall herbaceous plants (0.5–0.7 m) with shallow slow-flowing water.

## 1. Introduction

The white-winged flufftail (*Sarothrura ayresi*) is listed as critically endangered by the International Union for Conservation of Nature and faces a high risk of extinction [1–3]. The species' natural low-density occurrence, preference for wetland habitat, elusive behaviour and lack of clear auditory cues render it challenging to study.

The white-winged flufftail was first discovered breeding in the Ethiopian highlands at Sululta in the late 1990s, with a new breeding site found in the Berga wetlands in 1997 and a small population at Bilacha in 2005 [2]. Since there were no records of the birds breeding in South Africa, the general belief was that this cryptic species breeds in Ethiopia in the northern summer and migrates 4000 km south to South Africa during the austral summer [4].

A recent camera-trap study conducted in 2018 in the Middelpunt Wetland near the town of Dullstroom in South Africa confirmed the first breeding record of white-winged flufftail in the Southern Hemisphere [5]. This record established that, apart from the known breeding population in Ethiopia, a breeding population exists in the highlands of South Africa.

Through the combined use of camera traps and acoustic devices, the vocalization of the white-winged flufftail was characterized and unequivocally confirmed for the first time. The use of these devices further revealed a better understanding of white-winged flufftail behaviour and confirmed that the bird is a habitat specialist, preferring high-altitude wetlands with intact canopy and basal cover during the breeding season [5,6].

A similar unpublished study conducted in 2020 at Verloren Valei Nature Reserve revealed territorial displays and breeding behaviour of the white-winged flufftail. The birds are not resident at any of the known sites, sometimes departing after as little as six weeks when conditions become unfavourable at the onset of the dry season [7].

Data deficiencies, including a lack of detailed descriptions of the habitat requirements of the white-winged flufftail, have hindered past management strategies and conservation planning [8]. The plant community (species composition and structure) within the known habitat of the white-winged flufftail in South Africa has not been classified and described. It was, therefore, necessary to conduct a plant ecological study at a fine scale to help elucidate the habitat requirements of the species [9].

The aim of the study is to classify and describe the wetland vegetation of two known white-winged flufftail sites in Mpumalanga, South Africa, at the end of the breeding season. This information will assist practitioners in setting conservation targets to safeguard these irreplaceable habitats, for the benefit of these threatened birds and other species that rely on wetland ecosystems.

## 2. Methods

### 2.1. Study area

The white-winged flufftail was observed at two sites within Middelpunt Wetland in 2018 [5] and at three sites at Verloren Valei Nature Reserve in 2019/2020 (H Marais 2019, personal observation). These areas are located near the town of Dullstroom in Mpumalanga, a province of South Africa (figure 1).

The Verloren Vallei Nature Reserve is located approximately 12 km north of the town of Dullstroom and covers an area of 6055 ha (figure 1). The reserve contains 1041 ha of peat wetlands and is a Ramsar site [10]. Geologically, the reserve is underlain with quartzite, granite and shale of the Transvaal Sequence, Pretoria Group, Steenkampsberg Formation, with an intrusion of diabase layers throughout the formation [11]. Verloren Valei is a provincial nature reserve with an active management plan in place to maintain the natural ecological processes. According to this plan, wetlands are burned under cool and damp conditions and game species and numbers are managed to prevent overgrazing [12].

Middelpunt Wetland is located approximately 10 km south of Dullstroom and forms part of the recently declared Greater Lakenvlei Protected Environment (figure 1). This protected environment

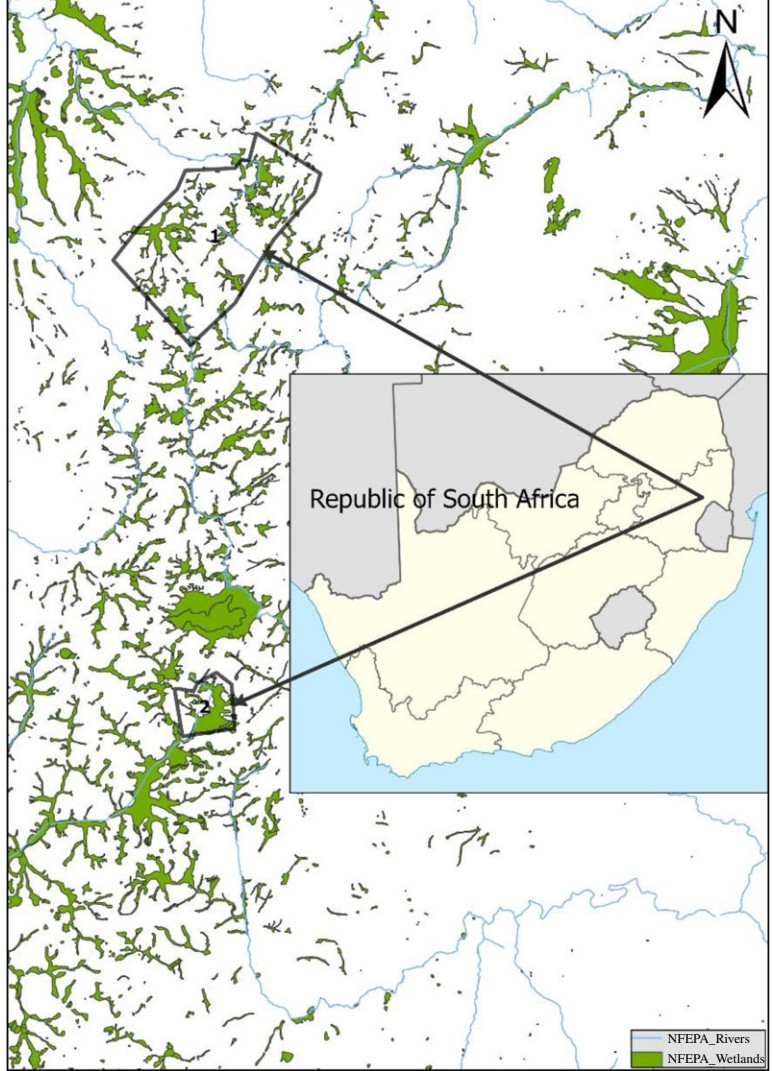

**Figure 1.** Location of Verloren Valei Nature Reserve (1) and Middelpunt Wetland (2).

is an important conservation area, proclaimed under the National Protected Areas Management Act [13], which gives private landowners the opportunity to conserve important biodiversity areas. The area comprises 9365 ha of which 1131 ha are peat wetlands, known as the Lakenvlei wetland system [10]. Middelpunt Wetland forms part of the upper reaches of this system and is located on a private farm.

The geology of the Lakenvlei catchment comprises quartzitic, cross-bedded sandstone of the Vryheid Formation, and hornfels with layers of silt and sandstone of the Vermont Formation in the south and Lakenvlei Formation quartzites in the west. Various northwest–southeast striking faults and north–south orientated diabase dykes transect the area, with diabase sills occurring in the north. The contact between a southwestern extension of the Lakenvlei quartzite and the hornfels of the Vermont Formation forms the key point of the main basin [14].

Although a management plan is in place for Middelpunt Wetland, it has not yet being fully implemented and accidental fires occurred in these wetlands on two occasions in 2019, during a very dry period. These fires were detrimental to the wetland.

Both study sites are located on the Steenkampsberg plateau at greater than 2000 m above sea level. They are located within the Highveld climate region and have a cool, moist climate, with severe frost occurring during the winter months. The average temperature ranges from −13°C to 29°C. The sites fall within the summer rainfall zone and have a mean annual rainfall of 840 mm [14]. The vegetation of the broader landscape is classified as Lydenburg Montane Grassland [15], with wetland ecosystems classified as Central Highveld Peat Wetlands [16].

## 2.2. Vegetation sampling

Field sampling of the vegetation was conducted in the areas where the birds were heard or seen during March 2020, shortly after the white-winged flufftail breeding season. A total of five sites were surveyed: three at Verloren Valei Nature Reserve and two at Middelpunt Wetland.

The study areas were visited prior to the surveys and delineated into homogeneous units using Google Earth images. A total of 12 sample plots (16 m$^2$) were placed at each site in a stratified random manner within the delineated areas of the wetland. Brown et al. [17], recommend that sample plot sizes for wetlands should range between 9 m$^2$ and 25 m$^2$. Sample plots of 4 × 4 m (16 m$^2$) were sufficient to classify the wetland plant communities at a fine scale.

In each sample plot, all plant species were recorded and the cover-abundance of each species was assessed using the modified Braun-Blanquet cover-abundance scale [18]. Environmental data such as wetland type, terrain units, topography, altitude, geology, soil characteristics, average water depth and average height of each species were collected. Plant height was determined in each sample plot by surveying 10 random points at 1 m intervals and measuring the inflorescence height of the nearest plant encountered ($n = 120$ per wetland surveyed). The wetland types were described by making use of hydrogeomorphological (HGM) units, as listed in the National Wetland Classification System (i.e. slope, depression, channelled-valley bottom, unchannelled-valley bottom [19]. The hydro-period and water depth were also noted. The plant species were identified using a wetland plant identification guide [20].

## 2.3. Data analysis

The vegetation data were captured into Excel and imported into the JUICE 7.1 software program [21] for analysis. A modified TWINSPAN classification [22] was performed to derive a first approximation of the vegetation units. TWINSPAN is a dichotomized ordination analysis for classification of vegetation into homogeneous groups. Pseudospecies cut levels were set at 0–5–15–25–50–75 [17]. The phi coefficient of association [23] was used to determine fidelity, a measure of species concentration in vegetation units [23], of each community. A phytosociological table was compiled by refining the table according to Braun-Blanquet procedures [24]. The diagnostic (species with high fidelity values), dominant (species with highest cover abundance) and constant species (number of relevés within which each species occurs), as statistically determined from the synoptic table, are listed for each sub-community. The lower threshold values were set at 60, 70 and 35 for fidelity, frequency and cover, respectively, and the upper thresholds at 80, 90 and 60. Plant community names were assigned in accordance with the recommendations of Brown et al. [17]. A Pareto chart was compiled in Excel to calculate the constancy distribution of plant species for all relevés to determine the species with the highest constancy values in all sub-communities. The plant species similarity between the sub-communities was measured using the Jaccard similarity index [25]. The shared species in each sub-community were listed in an Excel table and the similarity calculated according to the following formula:

$$\text{Jaccard index} = \frac{J}{A + B - J},$$

where $J$ = shared species and $A$ and $B$ are the species richness of the two samples being compared [25].

# 3. Results

## 3.1. Vegetation classification

The results of the vegetation classification of Verloren Valei Nature Reserve and Middelpunt Wetland are presented in table 1.

Both study sites are classified as belonging to the one main community, with three sub-communities (all references to species groups are reflected in table 1):

1. *Carex austro-africana–Cyperus denudatus* wetland
    1.1. *Carex austro-africana–Cyperus denudatus–Fuirena ciliaris* sub-community
    1.2. *Carex austro-africana–Cyperus denudatus–Carex cognata* sub-community
    1.3. *Carex austro-africana–Cyperus denudatus–Leersia hexandra* sub-community
1. *Carex austro-africana–Cyperus denudatus* wetland

**Table 1.** Phytosociological table of the study area indicating the different plant communities of Verloren Valei Nature Reserve and Middelpunt Wetland.

| Plant community number | | 1 | | |
|---|---|---|---|---|
| | 1.1 | 1.2 | | 1.3 |

Relevé number: 1  6  7  11  24  21  33  34  31  35  32  36  23  19  18  8  9  12  10  20  17  22  4  3  2  5  15  13  26  16  27  30  28  29  25  14  40  37  41  39  38  45  44  46  43  60  59  58  57  42  56  49  47  50  51  48  55  52  53  54

*Species group A (characteristic species for the Carex austro-africana–Cyperus denudatus wetland)*

- Cyperus denudatus
- Carex austro-africana
- Fuirena ciliaris
- Helichrysum mundtii
- Mentha aquatica
- Persicaria decipiens
- Senecio inornatus
- Carex cognata

*Species group B (characteristic species for the Carex austro-africana–Cyperus denudatus–Carex cognata sub-community)*

- Ascolepis capensis
- Schoenoplectus brachyceras
- Juncus oxycarpus
- Isolepis fluitans
- Lobelia flaccida
- Senecio oxyriifolius
- Watsonia bella

*Species group C (characteristic species for the Carex austro-africana–Cyperus denudatus–Leersia hexandra sub-community)*

- Leersia hexandra
- Arundinella nepalensis
- Juncus punctorius
- Persicaria lapathifolia*
- Phragmites australis
- Koeleria capensis
- Pseudognaphalium luteo-album

(Continued.)

**Table 1.** (*Continued.*)

| Plant community number | 1 | | | | | | | | | | | | | | | | | | | | | | | | | | | | | | | | | | | | | | | | | | | | | | | | | | | | | | | | | |
|---|---|---|---|---|---|---|---|---|---|---|---|---|---|---|---|---|---|---|---|---|---|---|---|---|---|---|---|---|---|---|---|---|---|---|---|---|---|---|---|---|---|---|---|---|---|---|---|---|---|---|---|---|---|---|---|---|---|---|
| | 1.1 | | | | | | | | | | | | | | | | | | | | | | 1.2 | | | | | | | | | | | | | | 1.3 | | | | | | | | | | | | | | | | | | | | | | |
| Relevé number | 1 | 6 | 7 | 11 | 24 | 21 | 33 | 34 | 31 | 35 | 32 | 36 | 23 | 19 | 18 | 9 | 8 | 12 | 10 | 20 | 17 | 22 | 4 | 3 | 2 | 5 | 15 | 13 | 26 | 16 | 27 | 30 | 28 | 29 | 25 | 14 | 40 | 37 | 41 | 39 | 38 | 45 | 44 | 46 | 43 | 60 | 59 | 58 | 57 | 42 | 56 | 49 | 47 | 50 | 51 | 48 | 55 | 52 | 5354 |
| *Pycnostachys reticulata* | . | . | . | . | r | . | . | . | . | . | . | . | . | . | . | . | . | . | . | . | . | . | . | . | . | . | . | . | . | + | . | . | . | . | . | . | . | . | r | . | . | . | . | . | . | . | . | . | r | . | r | . | . | . | . | . | . | . | .. |
| *Ranunculus multifidus* | . | . | . | . | . | . | . | . | . | . | . | . | . | . | . | . | . | . | . | . | . | . | . | . | . | . | . | . | . | . | . | . | . | . | . | . | + | + | r | + | . | . | . | . | 1 | . | . | . | . | + | . | . | + | . | . | 1 | . | . | .. |
| *Schoenoplectus brachyceras* | . | . | . | . | . | . | . | . | . | . | . | . | . | . | . | . | . | . | . | . | . | . | . | . | . | . | . | . | . | . | . | . | . | . | . | . | . | . | . | . | . | . | . | . | . | + | + | + | + | . | . | . | . | . | + | . | . | . | .. |
| *Senecio mundii* | . | . | . | . | . | . | . | . | . | . | . | . | . | . | . | . | . | . | . | . | . | . | . | . | . | . | . | . | . | . | . | . | . | . | . | . | r | . | r | . | r | r | . | + | . | . | . | . | . | . | . | . | . | . | . | . | + | r | .r |
| *Setaria sphacelata* | . | . | . | . | . | . | . | . | . | . | . | . | . | . | . | . | . | . | . | . | . | . | + | . | . | . | . | . | . | . | . | . | . | . | . | . | . | r | + | + | r | + | + | r | . | . | . | . | + | 1 | + | . | 1 | . | . | . | r | . | .. |
| *Typha capensis* | . | 5 | . | . | . | . | . | . | . | . | . | . | . | . | . | . | . | . | . | . | . | . | . | . | . | . | . | . | . | . | . | . | . | . | . | . | + | r | + | + | + | + | . | . | . | r | r | r | . | . | . | . | . | . | . | r | . | . | +. |
| *Verbena bonariensis* | . | . | . | . | . | . | . | . | . | . | . | . | . | . | . | . | . | . | . | . | . | . | . | . | . | . | . | . | . | . | . | . | . | . | . | . | . | . | . | . | r | . | . | . | . | . | . | . | . | r | . | . | r | . | . | r | . | . | .r |
| *Alepidea attenuata* | . | . | . | . | . | . | . | . | . | . | . | . | . | . | . | . | . | . | . | . | . | . | . | . | . | . | . | + | . | . | . | . | . | r | . | . | . | . | . | r | r | . | . | . | . | r | r | r | . | . | . | r | . | . | . | . | . | . | .. |
| *Berula erecta* | . | . | . | . | . | . | . | . | . | . | . | . | . | . | . | . | . | . | . | . | . | . | . | . | . | . | . | . | . | . | . | . | . | . | . | . | . | . | . | . | r | . | . | . | . | r | r | r | r | . | . | . | . | . | . | . | . | . | .. |
| *Calamagrostis epigejos* | . | . | . | . | . | . | . | . | . | . | . | . | . | . | . | . | . | . | . | . | . | . | . | . | . | . | . | . | . | . | . | . | . | . | . | . | . | r | . | . | . | . | . | r | . | . | . | . | r | . | r | . | r | . | . | . | . | . | .. |
| *Cephalaria oblongifolia* | . | . | . | . | . | . | . | . | . | . | . | . | . | . | . | . | . | . | . | . | . | . | . | . | . | . | . | . | . | . | . | . | . | . | . | . | . | . | r | . | r | . | . | . | r | . | . | . | . | r | . | r | r | . | r | r | . | . | .. |
| *Agrostis montevidense* | r | r | r | r | . | . | . | . | . | . | . | . | . | . | . | . | . | . | . | . | . | . | . | . | . | . | . | . | . | . | . | . | . | . | . | . | . | . | r | . | . | . | . | . | + | . | . | . | . | + | . | + | + | . | . | . | . | . | .. |

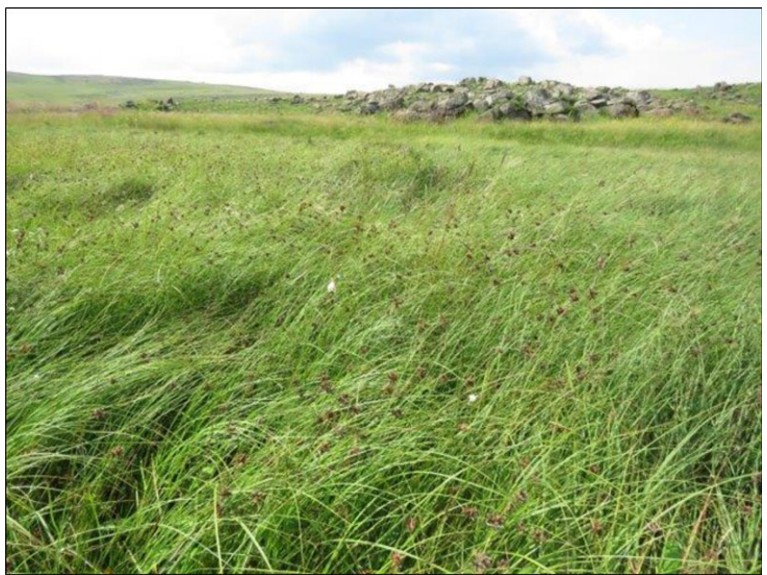

**Figure 2.** Photo indicating general appearance of the *Carex austro-africana–Cyperus denudatus–Fuirena ciliaris* sub-community.

This community is characterized by species from species group A. The vegetation is dominated by the sedges *Cyperus denudatus* (83% constancy) and *Carex austro-africana* (63% constancy), while the forbs *Persicaria decipiens* and *Senecio inornatus* (species group A) are prominent throughout the community, with a 70% and 60% constancy, respectively.

1.1. *Carex austro-africana–Cyperus denudatus–Fuirena ciliaris* sub-community

This sub-community is similar to the main community, but is characterized by the absence of species from species groups B and C. The vegetation is dominated by the sedge *Carex austro-africana*, while the sedges *Cyperus denudatus*, *Fuirena ciliaris* and the forb *Senecio inornatus* are locally prominent (figure 2).

From the synoptic table analysis, this community has the following diagnostic, constant and dominant species:

| | |
|---|---|
| diagnostic species | none |
| constant species | *Carex austro-africana* |
| dominant species | *Carex austro-africana, Typha capensis* |

1.2 *Carex austro-africana–Cyperus denudatus–Carex cognata* sub-community

This sub-community is characterized by species from species group B, of which the sedge *Ascolepis capensis* is the most prominent. The vegetation is dominated by the sedge *Carex cognata, Cyperus denudatus* and the forb *Fuirena ciliaris* (species group A) (figure 3).

From the synoptic table analysis, this community has the following diagnostic, constant and dominant species:

| | |
|---|---|
| diagnostic species | *Ascolepis capensis* |
| constant species | *Carex cognata, Cyperus denudatus, Persicaria decipiens* |
| dominant species | *Carex cognata, Fuirena ciliaris* |

1.3. *Carex austro-africana–Cyperus denudatus–Leersia hexandra* sub-community

This community is characterized by the presence of species from species group C. The grasses *Leersia hexandra* and *Arundinella nepalensis* (species group C) and the sedge *Cyperus denudatus* (species group A) dominate the vegetation (figure 4).

From the synoptic table analysis, this community has the following diagnostic, constant and dominant species:

| | |
|---|---|
| diagnostic species | *Leersia hexandra, Persicaria lapathifolia, Phragmites australis, Pseudognaphalium luteo-album* |
| constant species | *Cyperus denudatus, Persicaria decipiens, Senecio inornatus* |
| dominant species | *Arundinella nepalensis, Cyperus denudatus, Leersia hexandra* |

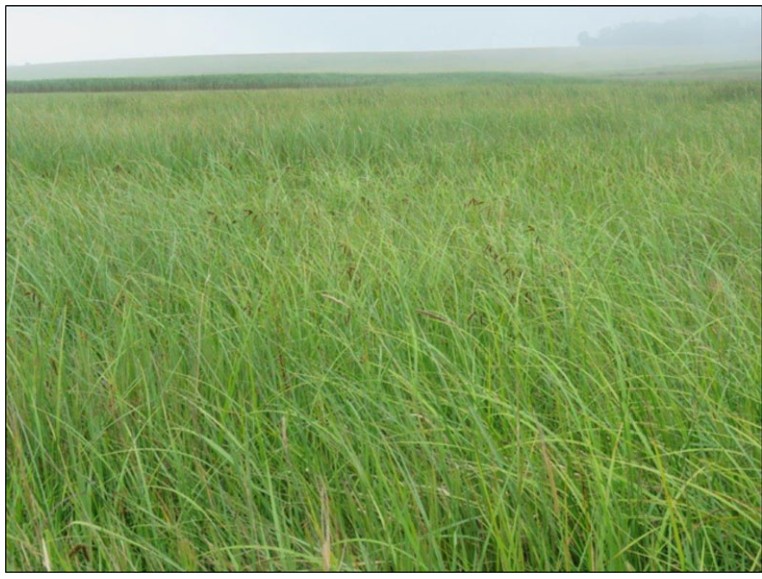

**Figure 3.** Photo indicating general appearance of the *Carex austro-africana–Cyperus denudatus–Carex cognata* sub-community.

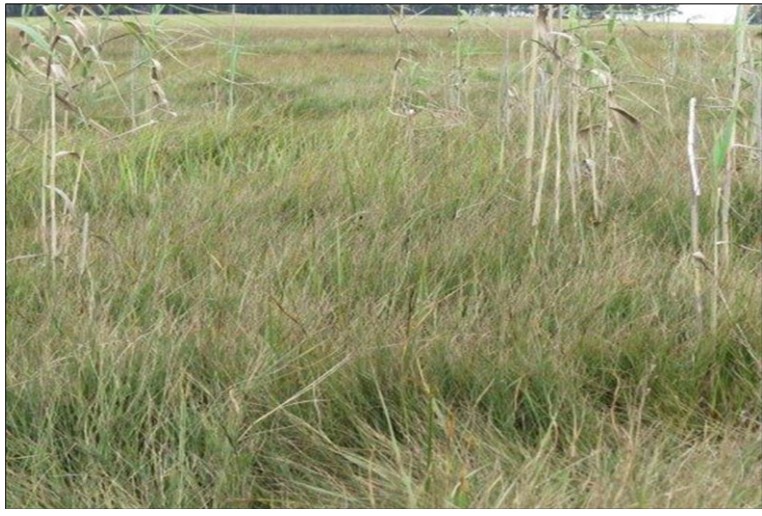

**Figure 4.** Photo indicating general appearance of the *Carex austro-africana–Cyperus denudatus–Leersia hexandra* sub-community.

## 3.2. Common species and similarity among sub-communities

The species with constancy values higher than 60% in all sub-communities are *Cyperus denudatus* (83%), *Persicaria decipiens* (70%), *Carex austro-africana* (63%) and *Senecio inornatus* (60%) (figure 5). These four species account for approximately 30% of all the plant species identified in the wetlands. A total of 11 species account for 70% of all species, namely *Cyperus denudatus, Persicaria decipiens, Carex austro-africana, Senecio inornatus, Leersia hexandra, Helichrysum mundtii, Mentha aquatica, Carex cognata, Arundinella nepalensis, Juncus punctorius* and *Fuirena ciliaris* (figure 5).

The species with high constancy values per sub-community were the sedges *Cyperus denudatus* and *Carex austro-africana*, with constancy values higher than 40% in each sub-community (figure 6). The forb *Senecio inornatus* also has a high constancy in all three sub-communities; however, it is more associated with sub-communities 1.2 and 1.3 (figure 4). The forb *Persicaria decipiens* is mostly associated with sub-communities 1.2 and 1.3, while the forb *Senecio inornatus* is mostly associated with sub-communities 1.1 and 1.3. The sedge *Carex cognata* and the grass *Ascolepis capensis* have high constancy in sub-community 1.2. The grasses *Leersia hexandra, Setaria sphacelata* and *Arundinella nepalensis*, the reed *Phragmites australis*, the forb *Typha capensis* and the alien weed *Persicaria lapathifolia* are mostly associated with sub-community 1.3 (figure 6).

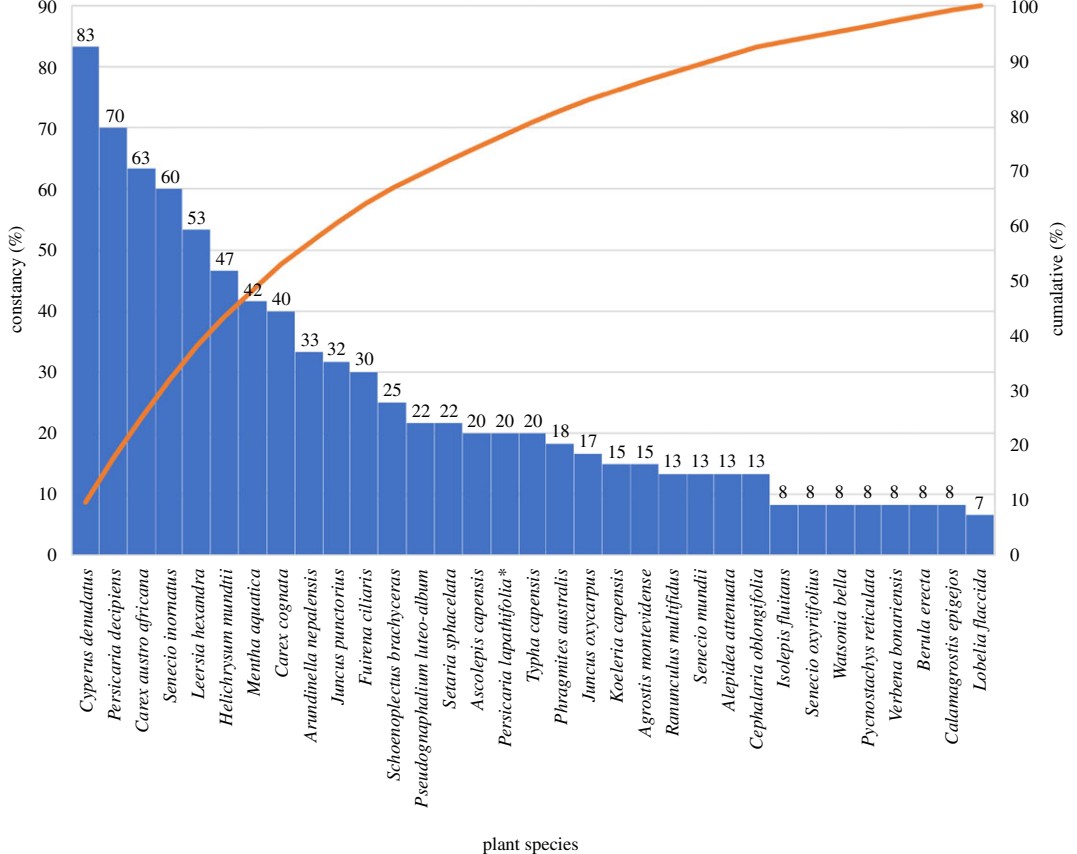

**Figure 5.** A Pareto chart indicating the constancy distribution of species for all the relevés in descending order (Orange line = cumulative line as % of the total).

The three sub-communities are fairly similar to each other in their species composition. Sub-community 1.1 shares a total of 22 species with sub-community 1.2 and has a 59% similarity, while sub-community 1.1 shares 18 species with sub-community 1.3, with a 46% similarity (table 2). There is less similarity between sub-community 1.2 and 1.3. If only the dominant and characteristic species are considered, sub-community 1.1 is 74% similar to sub-community 1.2 and 50% similar to sub-community 1.3, while sub-community 1.2 is 47% similar to sub-community 1.3.

## 3.3. Species traits and habitat characteristics

The dominant species ranged in height between 0.45 and 1.22 m, with *Carex austro-africana* being the tallest and *Ascolepis capensis* being the shortest (table 3). The average vegetation height of the most dominant species was 0.67 m (±0.3 s.d.).

All five of the most dominant species are perennial, with rhizomes and/or rootstocks, and prefer shallow to marshy habitats (table 3). The average water depth of the wetlands during the vegetation sampling was 0.47 and 0.85 m for Middelpunt Wetland and Verloren Valei Nature Reserve, respectively.

Canopy cover as estimated during the plot surveys using the modified Braun-Blanquet cover abundance scale ranged between 90% and 100%. The vegetation was dense but had small gaps between the different plant species that allowed easy access to small birds for foraging and shelter.

## 3.4. Plant families

A total of 18 plant families are represented at the two study sites. The Cyperaceae is the top-ranked family, with the most species, followed by the Asteraceae and the Poaceae, for all the sub-communities together (table 4). These three families are also ranked the top three families in each of the three sub-communities, although the Asteraceae and the Poaceae are the top two in sub-community 1.3 and the Cyperaceae is the top-ranked family in sub-communities 1.1 and 1.2 (table 4). The top three families represent 63%, 56% and

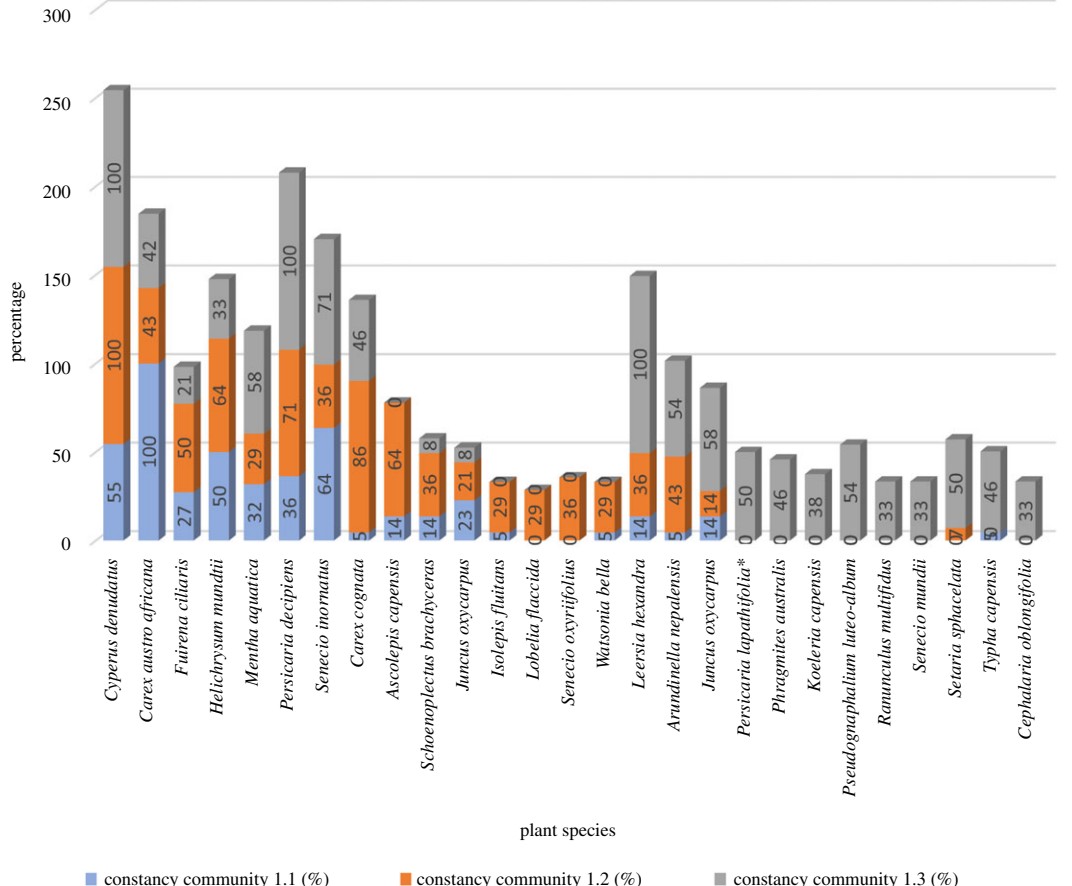

**Figure 6.** Constancy values of each species per sub-community.

**Table 2.** Jaccard index values for plant species similarity for three sub-communities of the study area.

|  | sub-community 1.1 | sub-community 1.2 | sub-community 1.3 |
|---|---|---|---|
| shared species |  |  |  |
| sub-community 1.1 |  |  |  |
| sub-community 1.2 | 22 |  |  |
| sub-community 1.3 | 18 | 18 |  |
| Jaccard similarity values |  |  |  |
| sub-community 1.1 |  |  |  |
| sub-community 1.2 | 0.59 |  |  |
| sub-community 1.3 | 0.46 | 0.38 |  |

63% of all the species present in sub-communities 1.1, 1.2 and 1.3, respectively. Prominent families in all three sub-communities include the Juncaceae and the Lamiaceae (table 3).

## 4. Discussion

Physical observations as well as audio recordings of the white-winged flufftail were documented at Middelpunt Wetland using camera trap and acoustic survey designs [5,6] and Verloren Valei Nature Reserve. The substrate of the wetlands at these sites is characterized by a substantial accumulation of organic matter and they are classified as peatlands [26]. It is important that the plant communities of an area are classified, described and interpreted as natural resources [17], since it would facilitate the

**Table 3.** Traits and heights of characteristic plant species found in the plant communities used by white-winged flufftail at Middelpunt Wetland and Verloren Valei Nature Reserve.

| species | average height (mm) | s.d. | life cycle | root system | habitat |
|---|---|---|---|---|---|
| **Cyperus denudatus** | 450 | ±30 | perennial | rhizomes | |
| **Carex austro-africana** | 1225 | ±105 | perennial | rhizome | shallow slow-flowing water |
| Fuirena ciliaris | 730 | ±20 | annual | thickened stem | swampy grassland |
| Helichrysum mundtii | 706 | ±122 | perennial | rootstock | moist areas |
| Mentha aquatica | 409 | ±25 | perennial | rhizomes | moist areas |
| **Persicaria decipiens** | 378 | ±58 | annual/ perennial | rhizomes/stolons | water and wet soil |
| **Senecio inornatus** | 852 | ±147 | perennial | rootstock | marshy patches |
| **Carex cognata** | 723 | ±41 | perennial | rhizomes/ rootstock/ stolons | marshy, permanently wet areas |
| **Ascolepis capensis** | 397 | ±37 | perennial | rhizome | damp spots |

**Table 4.** Ranking (R) of the top eight plant families, number of species (Tot sp.), and percentage of total species (% Sp.) for all plant sub-communities used by white-winged flufftail at Middelpunt Wetland and Verloren Valei Nature Reserve, Mpumalanga, South Africa.

| family | all communities | | | community 1 | | | community 2 | | | community 3 | | |
|---|---|---|---|---|---|---|---|---|---|---|---|---|
| | R | Tot sp. | % Sp. | R | Tot sp. | % Sp. | R | Tot sp. | % Sp. | R | Tot sp. | % Sp. |
| Cyperaceae | 1 | 9 | 20 | 1 | 8 | 33.3 | 1 | 8 | 25.0 | 3 | 5 | 16.7 |
| Asteraceae | 2 | 8 | 17.8 | 2 | 4 | 16.7 | 2 | 6 | 18.8 | 1 | 7 | 23.3 |
| Poaceae | 2 | 8 | 17.8 | 3 | 3 | 12.5 | 3 | 4 | 12.5 | 1 | 7 | 23.3 |
| Apiaceae | 4 | 2 | 4.4 | | | | | | | | | |
| Iridaceae | 4 | 2 | 4.4 | | | | 4 | 2 | 6.3 | 4 | 2 | 6.3 |
| Juncaceae | 4 | 2 | 4.4 | 4 | 2 | 8.3 | 4 | 2 | 6.3 | 4 | 2 | 6.3 |
| Lamiaceae | 4 | 2 | 4.4 | 4 | 2 | 8.3 | 4 | 2 | 6.3 | 4 | 2 | 6.3 |
| Polygonaceae | 4 | 2 | 4.4 | | | | | | | | | |

implementation of scientifically justifiable management actions [27] as well as assist in our understanding of the ecology of various bird species.

All three sub-communities identified share the same dominant species and only differ in terms of the presence of plant species that have a relatively low canopy cover and a moderate constancy. This, therefore, indicates broadly similar habitat conditions existing between the study areas where the white-winged flufftail was observed.

A total number of 45 different plant species that represent 38 genera and 18 plant families were identified. The most prominent plant families present were the Cyperaceae, the Asteraceae and the Poaceae. This matches the findings of Allan *et al.* [28] in the Berga marsh in Ethiopia, where the white-winged flufftail also occurs.

The study area has affinity with the *Phragmites australis–Carex austro-africana* community, the *Leersia hexandra–Arundinella nepalensis* community and the *Carex austro-africana–Schoenoplectus brachyceras* community of the Hlatikulu vlei, as described by Guthrie [29] in the foothills of the southern Drakensberg. Sieben *et al.* [30] has also described wetland communities that show an affinity to the study area, namely the *Cyperus solidus–Leersia hexandra* community in Maputoland, the *Leersia hexandra*

(a)    (b)

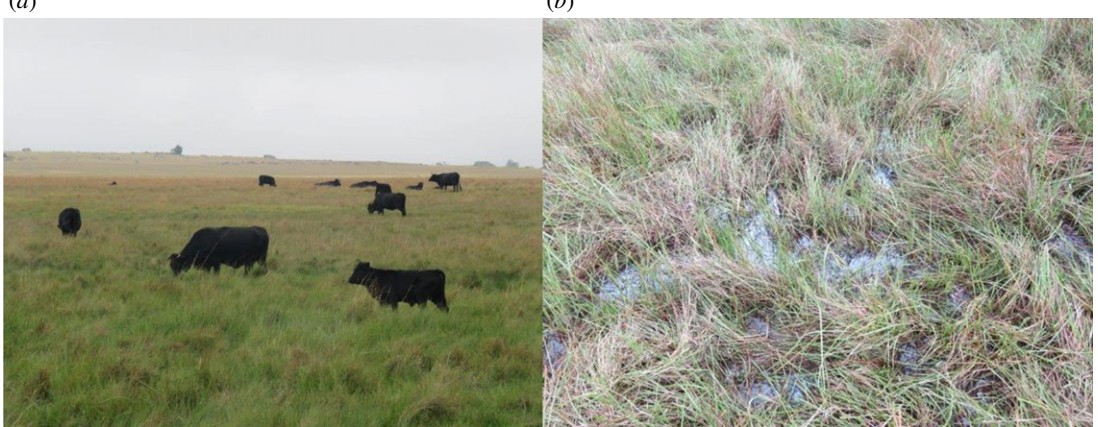

**Figure 7.** (a) Cattle grazing intensively at Middelpunt Wetland, with (b) resultant trampling of the sensitive wetland vegetation (picture: A. J. Marais, Middelpunt Wetland, 2021).

community of temperate grassy wetlands in the northern parts of South Africa and the *Carex austro-africana* community along the eastern escarpment of the Drakensberg. These sites, along with the Steenkampsberg, where the study sites were located, form part of the same discontinuous escarpment that spans the length of southern Africa. It, therefore, seems that similar habitats are available in other parts of the country, which could be monitored for the presence of the white-winged flufftail. Although Mendelsohn *et al.* [4] was unsuccessful in finding these birds in KwaZulu-Natal, passive monitoring techniques have since been developed that could increase detection rates [6].

The wetland communities of Verloren Valei Nature Reserve are characterized by the dominance of the sedges *Carex austro-africana* and *Cyperus denudatus*, with the sedge *Carex cognata* and the forb *Senecio inornatus* prominent locally. Although similar in terms of the dominant vegetation of Verloren Valei Nature Reserve, the vegetation of Middelpunt Wetland is characterized by a larger component of grass species, especially the co-dominance of the grass *Leersia hexandra* and the prominence of the grass *Arundinella nepalensis*. The predominance of these two species can probably be attributed to drier conditions prevailing at Middelpunt Wetland. The prominence of the grass *Leersia hexandra* indicates intermediate conditions in terms of soil wetness [30], which is evident from recorded water levels. This was also found by Guthrie [29], who described the soils where the grasses *Leersia hexandra* and *Arundinella nepalensis* occurred as permanently waterlogged, but with little standing water. These conditions at Middelpunt Wetland are probably promoted by regular fires, grazing and trampling by cattle (figure 7). Excessive grazing throughout the year and unplanned fires remove vegetation cover, which assists in water evaporation, creating drier conditions [31].

Areas become unsuitable for the white-winged flufftail when vegetation structure is degraded [28]. The effect of cattle grazing is also evident in the dispersal of the declared alien invader weed *Persicaria lapathifolia*, which easily establishes in disturbed areas where moist conditions prevail [32]. This weed, which is prominent within community 1.3 (Middelpunt Wetland), is normally introduced into peatlands through drainage water from adjacent cultivated lands or areas where cattle are fed. Leyer [33] found surface water to be the major dispersal strategy of pioneer species such as *Persicaria lapathifolia* into water systems where connectivity exists, as is the case with Middelpunt Wetland. Based on the vegetation composition and characteristic species of Middelpunt Wetland, it seems as though sections of the wetland are drier and more affected by grazing than others. Various cattle paths exist within sections of the wetland where grazing was evident (H Marais 2019, personal observation). These cattle paths assist in concentrating water flow into the paths owing to lateral drainage from the adjacent wetland areas, which causes further drying of this wetland. This was also found by Du Preez & Brown [34] in the high-altitude peatlands of Lesotho. Regular grazing by cattle flattens wetland vegetation, causing degradation of the habitat (e.g. loss in vegetation cover, decline in species diversity, soil erosion), while the presence of cattle can also act as a deterrent and lead to the birds relocating to other suitable areas [4]. This was also found by Allan *et al.* [28] in the Berga marsh in Ethiopia, where the extensive grazing by domestic animals is causing large-scale degradation of the habitat. No white-winged flufftails were found in the areas where these anthropogenic activities occurred [28].

Based on detailed vegetation surveys of areas in proximity to those where the white-winged flufftail was observed and/or recorded, a total of 11 wetland plant species accounted for 70% of all species

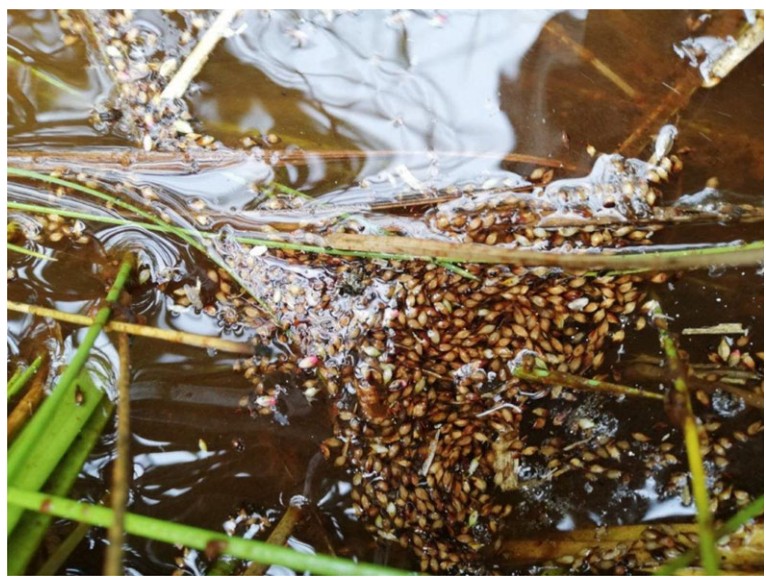

**Figure 8.** Mature seeds of *Carex cognata* and *Carex austro-africana* floating on the water surface provide easily available food for wading birds (picture: A. J. Marais, Verloren Valei nature Reserve, 2020).

surveyed at both wetlands. The most conspicuous plant species present at both wetlands were the sedges *Cyperus denudatus* and *Carex austro-africana*, and the forbs *Persicaria decipiens* and *Senecio inornatus*. These species occur in permanently wet areas with mostly shallow, slow-flowing water. The sedge *Cyperus denudatus* and the grass *Arundinella nepalensis* are indicators of permanently wet areas [30].

Vegetation cover at the study sites was high, estimated at an average of 90–100%. The average height of the vegetation at the five study sites was measured at 0.58 m (±0.06 s.d.), but varied between 0.48 and 0.64 m. *Carex austro-africana* generally remains lower in growth structure than other *Carex* species [30]. However, it was measured to be up to 1.4 m tall (average 1.2 ± 0.11 s.d.). According to Davies *et al.* [9] a white-winged flufftail was flushed from Middelpunt Wetland from 0.55 m high vegetation dominated by the sedges *Pycreus macranthus*, *Fuirena thunbergii* and *Leersia hexandra*, while another bird was flushed from the wetlands at Verloren Valei Nature Reserve in an area dominated by *Carex* and *Cyperus* sedges between 1 and 1.5 m tall. The nests of these birds were found within vegetation ranging in height between 0.2 and 0.4 m in the wetlands of Ethiopia [5]. It, therefore, seems that the birds occur in wetland areas that have a generally lower height structure (knee-height), ranging between 0.2 and 0.66 m [1,9], but they also use *Carex*-dominated patches of up to 1.5 m tall. According to Colyn *et al.* [5], the white-winged flufftail prefers shallowly flooded areas with water less than 0.1 m deep. The birds flushed by Davies *et al.* [9] in Middelpunt and Verloren Valei wetlands were from areas where the water depth ranged between 0.5 and 0.2 m. The sedge *Carex austro-africana* normally grows in water not deeper than 0.5 m [35] and it, therefore, seems that the white-winged flufftails use shallow water areas ranging in depth of between 0.1 and 0.5 m.

Whereas some observations of the white-winged flufftail indicate that they flee into *Phragmites* and *Typha* reedbeds, no bird has been flushed from these areas [4,9]. All flushed records have been of birds in adjacent sedge-dominated wetland areas. The results of this study also indicate that reedbeds are not preferred habitat, with none of these species dominant in the plant communities identified where the white-winged flufftail has been observed at Verloren Valei Nature Reserve and Middelpunt Wetland.

Most of the plant species within the habitat used by white-winged flufftail are perennial, with rhizomes or rootstocks that enable them to successfully reproduce in saturated soil. Mofutsanyana [36] found that plants with rhizomes are able to establish in waterlogged soil and outcompete other plants that do not have similar traits. The dominant species in the study areas are obligate or facultative wetland species. Obligate wetland species (e.g. *Cyperus denudatus*, *Carex austro-africana*, *Carex cognata*, *Persicaria decipiens*, *Leersia hexandra*, *Arundinella nepalensis* and *Juncus punctorius*) have higher root aerenchymatous volume than facultative wetland species [37]. Should soil conditions become drier, these plants would decrease in number and density, since they are non-adaptive to dry conditions owing to their higher root porosity and resultant lower mechanical strength [38]. As stated earlier, continued grazing of wetlands can increase the dryness of these wetlands, which would lead to obligate wetland species decreasing in number with resultant degradation of the white-winged flufftail habitat.

Based on the results of this study, it is hypothesized that the dense vegetation and high canopy cover of *Carex* and *Cyperus* plant communities provide suitable habitat for the white-winged flufftail. Taylor & van Perlo [39] found the species to breed in high-altitude marshes with dense vegetation that is dominated by sedges and grasses. The dense vegetation and permanently wet conditions provide an ideal habitat for insects and crustaceans, on which the birds feed [40]. Both *Carex cognata* and *Carex austro-africana* have inflorescences that produce numerous seeds, which also provide a potential food source for these birds during the breeding season. When the seeds mature, they drop down onto the water and become easily available to wading birds, such as the white-winged flufftail (figure 8).

Allan *et al.* [28] noted similar habitat in the Berga marsh in Ethiopia, where the white-winged flufftail was found to be breeding. Although not dominated by the same plant species as in the study sites, the wetland vegetation structure and water depth in the Berga marsh are similar to those of the wetlands studied.

## 5. Conclusion

The white-winged flufftail is a small, elusive bird that is known to occur only in the high-altitude wetlands of South Africa and Ethiopia [41]. This detailed habitat study shows that the high-altitude peatlands in Mpumalanga, South Africa where the birds were observed are dominated by the sedges *Carex austro-africana* and *Cyperus denudatus*, with the grasses *Leersia hexandra* and *Arundinella nepalensis* co-dominant. These areas are permanently wet with shallow slow-flowing water that ranges in depth between 0.1 and 0.5 m. The medium-tall vegetation (0.5–0.7 m) is dense with a high canopy cover (90–100%) that provides shelter and foraging opportunities to the birds. The dominant plant species within these habitats are perennial and have specific traits that enable them to survive in these wet conditions. The vegetation structure and water depth of the habitat in the study area is similar to that of wetland habitats in Ethiopia, where the birds are also found. Wetland areas with similar vegetation composition and structure within the high-altitude areas of the Mpumalanga Province could provide suitable habitat for the white-winged flufftail and should be identified and monitored for their presence.

It is recommended that a detailed management and monitoring plan should be developed for these wetlands, taking grazing by domestic animals into consideration, to ensure the sustainable use of wetlands on farmlands while protecting the habitat from degradation. Grazing in these peatlands during the breeding season could deter the birds from breeding while incorrect stocking rates and continuous grazing within these habitats could change its vegetation structure and composition making it unsuitable for the white-winged flufftail.

Data accessibility. Data added as an extra file in Excel. All data available from the Dryad Digital Repository: https://doi.org/10.5061/dryad.m0cfxpp4n [42].

Authors' contributions. A.J.M. conceptualized the project and collected the field data and did initial data analysis and assisted with the manuscript writing. L.R.B. provided guidance in the project design and vegetation surveys, did the final data analyses and wrote large sections of the manuscript. H.A.S.-R. and K.L. provided comments and recommendations and assisted with detailed information on the white-winged flufftail.

Competing interests. At the time of writing and submission of the manuscript, L.R.B. as an Editorial Board member of the Royal Society Open Science, had no involvement in the review or assessment of the paper.

Funding. We received no funding for this study.

Acknowledgements. David Allan and Warwick Tarboton for their comments on the manuscript. The Mpumalanga Tourism and Parks Agency for supporting the study. The Dullstroom Trout Farm for permitting the study.

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
