## [Peer Review File · Royal Society Open Science]

Review History

RSOS-211482.R0 (Original submission)

Review form: Reviewer 1

Is the manuscript scientifically sound in its present form?

Yes

Are the interpretations and conclusions justified by the results?

Yes

Is the language acceptable?

Yes

Do you have any ethical concerns with this paper?

No

Have you any concerns about statistical analyses in this paper?

No

Recommendation?

Accept with minor revision (please list in comments)

Comments to the Author(s)

Your paper describing the species composition and structure of plant communities, and their habitat suitability for the rare White-winged Flufftail, at two high-elevation wetlands in Mpumalanga, South Africa, aims to fill an important gap in understanding what habitats currently best support bird foraging and breeding, and how to manage such habitats.

Your study achieves its objectives by providing a classification and description of the three vegetation sub-communities and what species characterise and distinguish the floristic assemblages from one another. A brief description of the structure (plant height) also aids in interpreting the current condition of the wetlands for White-winged flufftails.

The study is scientifically sound, based on appropriate vegetation sampling and community classification methodology. The manuscript is generally well-written and logical, providing an introductory context and stating the objectives of the research. The study sites and methodology used are clearly and concisely described. The results are adequately stated, illustrated with appropriate tables and graphs. The findings are discussed in the light of other studies and with respect to the biology and ecology of the bird species of interest. There is some repetition of introductory material in the Discussion. The Conclusions, however, are not cogent and concise; they appear to just extend the discussion rather than just stating the important conclusions that could be drawn from the study, especially regarding flufftail habitat suitability. I have included some questions of clarity and suggestions for improving the manuscripts annotated in the attached PDF.

Review form: Reviewer 2**Is the manuscript scientifically sound in its present form?**

Yes

Are the interpretations and conclusions justified by the results?

Yes

Is the language acceptable?

Yes

Do you have any ethical concerns with this paper?

No

Have you any concerns about statistical analyses in this paper?

Yes

Recommendation?

Accept with minor revision (please list in comments)

Comments to the Author(s)

None

Decision letter (RSOS-211482.R0)

Dear Professor Brown

On behalf of the Editors, we are pleased to inform you that your Manuscript RSOS-211482 "A vegetation classification and description of White-winged Flufftail (*Sarothrura ayresi*) habitat at selected high-altitude peatlands in South Africa." has been accepted for publication in Royal Society Open Science subject to minor revision in accordance with the referees' reports. Please find the referees' comments along with any feedback from the Editors below my signature.

Please submit your revised manuscript and required files (see below) no later than 7 days from today's (ie 11-Oct-2021) date. Note: the ScholarOne system will 'lock' if submission of the revision is attempted 7 or more days after the deadline. If you do not think you will be able to meet this deadline please contact the editorial office immediately.

on behalf of Dr Christie Bahlai (Associate Editor) and Pete Smith (Subject Editor)
openscience@royalsociety.org

Associate Editor Comments to Author (Dr Christie Bahlai):
Comments to the Author:

We have now received two reviews, and reviewers agree that the study is scientifically sound and well-written. Both reviewers have provided line-by-line annotations in an attached PDF, mainly points for clarification, so please ensure that in your revision, you address each of these comments. Reviewer 1 has also provided additional comments directly through the editorial system. I think the reviewers have done a nice job pointing out most issues, but I would like to reiterate Reviewer 1's comments- the conclusion repeats a lot of introductory material, so I would like to see that section streamlined, in particular.

Additionally, I'll admit I had to look up the word "physiognomy" (P10 L42) because I'd not seen it used in this context. It seems to be used more commonly to describe the study of human facial

features rather than the physical features of a plant habitat in many English dialects. To ensure broader readability it should be replaced with more universal terminology.

Reviewer comments to Author:

Reviewer: 1

Comments to the Author(s)

Your paper describing the species composition and structure of plant communities, and their habitat suitability for the rare White-winged Flufftail, at two high-elevation wetlands in Mpumalanga, South Africa, aims to fill an important gap in understanding what habitats currently best support bird foraging and breeding, and how to manage such habitats.

Your study achieves its objectives by providing a classification and description of the three vegetation sub-communities and what species characterise and distinguish the floristic assemblages from one another. A brief description of the structure (plant height) also aids in interpreting the current condition of the wetlands for White-winged flufftails.

The study is scientifically sound, based on appropriate vegetation sampling and community classification methodology. The manuscript is generally well-written and logical, providing an introductory context and stating the objectives of the research. The study sites and methodology used are clearly and concisely described. The results are adequately stated, illustrated with appropriate tables and graphs. The findings are discussed in the light of other studies and with respect to the biology and ecology of the bird species of interest. There is some repetition of introductory material in the Discussion. The Conclusions, however, are not cogent and concise; they appear to just extend the discussion rather than just stating the important conclusions that could be drawn from the study, especially regarding flufftail habitat suitability.

I have included some questions of clarity and suggestions for improving the manuscripts annotated in the attached PDF (White-winged Flufftail habitats_RSOS-211482_Comments.pdf)

Reviewer: 2

Comments to the Author(s)

None (attachment: RSOS-211482_Proof_hi 20211004.pdf)

===PREPARING YOUR MANUSCRIPT===

While not essential, it will speed up the preparation of your manuscript proof if you format your references/bibliography in Vancouver style (please see

<https://royalsociety.org/journals/authors/author-guidelines/#formatting>). You should include DOIs for as many of the references as possible.

===PREPARING YOUR REVISION IN SCHOLARONE===

<https://royalsociety.org/journals/authors/author-guidelines/#data>. You should ensure that you cite the dataset in your reference list. If you have deposited data etc in the Dryad repository,

please only include the 'For publication' link at this stage. You should remove the 'For review' link.

Author's Response to Decision Letter for (RSOS-211482.R0)

See Appendix A.

Decision letter (RSOS-211482.R1)

Dear Professor Brown,

I am pleased to inform you that your manuscript entitled "A vegetation classification and description of White-winged Flufftail (*Sarothrura ayresi*) habitat at selected high-altitude peatlands in South Africa." is now accepted for publication in Royal Society Open Science.

You can expect to receive a proof of your article in the near future. Please contact the editorial office (openscience@royalsociety.org) and the production office (openscience_proofs@royalsociety.org) to let us know if you are likely to be away from e-mail contact -- if you are going to be away, please nominate a co-author (if available) to manage the proofing process, and ensure they are copied into your email to the journal. Due to rapid

publication and an extremely tight schedule, if comments are not received, your paper may experience a delay in publication.

on behalf of Dr Christie Bahlai (Associate Editor) and Pete Smith (Subject Editor)
openscience@royalsociety.org

Appendix A

2021-10-12

Enquiries: Prof. L R BROWN
Telephone: (011) 471-2339
email: lrbrown@unisa.ac.za

Dr Christie Bahlai
Associate Editor
Royal Society Open Science

RESPONSE TO REVIEWERS' COMMENTS: A vegetation classification and description of White-winged Flufftail (*Sarothrura ayresi*) habitat at selected high-altitude peatlands in South Africa. (Marais, AJ, Lloyd K, Smit-Robinson, HA & Brown LR).

Dear Dr Bahlai

Thank you for the positive comments and recommendations aimed at improving the manuscript. We have gone through the manuscript and affected most of the changes as recommended by the reviewers. We have also shortened and streamlined the conclusion as requested and removed all repetitive sections. The changes as indicated by the reviewers are indicated in the table below:

Reviewer 1

Reviewer's comments	Affected	Authors response
P1. Conclusions regarding the suitability of the composition and structure of the three plant sub-communities for white-winged flufftails, and any recommended vegetation management practices to maintain or improve these habitats?	✓	The habitats were studied since they are currently suitable for the WWF. Thus, the paper deals with the description and composition of the habitat. We therefore feel that the abstract is correct
P4. Clarify your height measurement. Mean of how many measurements? Height of tallest leaf, leaf table height, inflorescence height?	✓	Agree. A sentence has been included clarifying the height measurements
P7. ... are fairly similar to each other in their species composition.	✓	Agree with changing of wording and have affected change
P8. How dense and complete was the horizontal vegetation cover, with any large or small gaps of bare area which could influence bird foraging, breeding, and shelter from predators?	✓	Although it is mentioned later on in the manuscript, we have included such information and agree that it fits into this section. We did not determine density, but could make a broad statement based on observations in the field
P9. This paragraph seems to reiterate some of the information provided in the Introduction.. Rather indicate your main findings and then continue to discuss them.	✓	We agree with the comment and have changed the paragraph to address the comment.

P9. This? The predominance of these two species...	✓	Affected wording change.
P9. At all times or seasonally?	✓	Have provided the necessary information
P9. How intense and continuous is the stocking that causes such trampling (throughout the year or only at times) and does it coincide with breeding when nests and eggs could be damaged?	✓	Have changed to sentence to address the question.
P10. Enlarge on what exactly you mean by degradation.	✓	Have changed sentence to address comment.
P10. Therefore flufftails prefer the wettest areas?	✓	Yes, we have addressed it in the conclusion.
P11. Possible consequences of such decreases for flufftail habitat?	✓	Have changed sentence to address comment
P11. dense [with wet areas for crustaceans]?	✓	Agree and have affected change
P11. ... not dominated by the same plant species...	✓	Affected word changes as recommended
P11. Avoid reiterating introductory statements - rather provide conclusions about your study and how they contribute to the important knowledge gap identified in the Introduction, and what is still unknown.	✓	Agree with comments and have affected changes to prevent duplication
P11. This [new] information appears out of any context or clear links to findings and other conclusions.	✓	Agree and have removed the section
P11. What intensity and timing of stocking / grazing would be benign [or even beneficial] for birds?	✓	Have addressed the comment in the text
P22. Perhaps label the second Y-axis.	✓	Have labelled the second axis as requested.

Reviewer 2

Reviewer's comments	Affected	Authors response
P3. plant species composition and not vegetation type	✓	Affected change
P4. Who was the observer?	✗	It is normally standard practice to indicate "pers. obs." without indicating the person since it normally refers to the 1 st author.

Various pages “plant communities”	✓	We agree with the comment and have changed all text stating “vegetation communities” to “plant communities”
P9. composition		We did not insert the term “composition” since we do not refer to the general composition, but to specific species that have low canopy cover.
Various pages. “soil”	✓	We affected three changes changing the word “soils’ to “soil”
P11. Maybe I have missed the physiognomy but alternative could be plant species composition and vegetation structure. For me physiognomy also include when it flower and seed as well as the vegetation structure.	✓	Yes agree that it is confusing and have changed to “vegetation structure”
P11. vegetation	✓	Have affected the changes as recommended.
P17. which locality?	✗	The figure is labelled correctly and states that it illustrates the Carex austro africana – Cyperus denudatus – Fuirena ciliaris sub-community. Thus no change needed
P22. Figure 6. Difficult to read the plant species names and X should be named Plant species as well as the right Y side name?	✓	Although it is the PDF version, we have addressed the comment by providing a higher resolution figure
P22, Figure 6. Name X and Y axis	✓	Thank you. Also noticed by reviewer 1 and have corrected it

I hope you find all in order.

Kind regards

PROFESSOR LR BROWN
APPLIED BEHAVIOURAL ECOLOGY & ECOSYSTEM RESEARCH UNIT
University of South Africa